# Three-Dimensional Bioprinting with Alginate by Freeform Reversible Embedding of Suspended Hydrogels with Tunable Physical Properties and Cell Proliferation

**DOI:** 10.3390/bioengineering9120807

**Published:** 2022-12-15

**Authors:** Yuanjia Zhu, Charles J. Stark, Sarah Madira, Sidarth Ethiraj, Akshay Venkatesh, Shreya Anilkumar, Jinsuh Jung, Seunghyun Lee, Catherine A. Wu, Sabrina K. Walsh, Gabriel A. Stankovich, Yi-Ping Joseph Woo

**Affiliations:** 1Department of Cardiothoracic Surgery, Stanford University, Stanford, CA 94305, USA; 2Department of Bioengineering, Stanford University, Stanford, CA 94305, USA

**Keywords:** 3D bioprinting, FRESH, hydrogel, RGD, alginate, tunable, physical property

## Abstract

Extrusion-based three-dimensional (3D) bioprinting is an emerging technology that allows for rapid bio-fabrication of scaffolds with live cells. Alginate is a soft biomaterial that has been studied extensively as a bio-ink to support cell growth in 3D constructs. However, native alginate is a bio-inert material that requires modifications to allow for cell adhesion and cell growth. Cells grown in modified alginates with the RGD (arginine-glycine-aspartate) motif, a naturally existing tripeptide sequence that is crucial to cell adhesion and proliferation, demonstrate enhanced cell adhesion, spreading, and differentiation. Recently, the bioprinting technique using freeform reversible embedding of suspended hydrogels (FRESH) has revolutionized 3D bioprinting, enabling the use of soft bio-inks that would otherwise collapse in air. However, the printability of RGD-modified alginates using the FRESH technique has not been evaluated. The associated physical properties and bioactivity of 3D bio-printed alginates after RGD modification remains unclear. In this study, we characterized the physical properties, printability, and cellular proliferation of native and RGD-modified alginate after extrusion-based 3D bioprinting in FRESH. We demonstrated tunable physical properties of native and RGD-modified alginates after FRESH 3D bioprinting. Sodium alginate with RGD modification, especially at a high concentration, was associated with greatly improved cell viability and integrin clustering, which further enhanced cell proliferation.

## 1. Introduction

Three-dimensional (3D) bioprinting is an emerging technology used to rapidly fabricate engineered tissues in vitro using biomaterials combined with live cells and growth factors [1,2,3,4]. Extrusion-based bioprinting works by depositing cell-laden bio-inks onto printing platforms in a layer-by-layer fashion to generate 3D constructs [5]. Biocompatibility, printability, and mechanical properties of the bio-ink are important factors to consider when selecting a suitable bio-ink for extrusion-based 3D bioprinting. Alginate is a soft biomaterial that has been frequently used as a bio-ink to support cell growth in 3D as a matrix scaffold due to its biocompatibility and tunable physical properties [3,6,7,8,9,10]. Although 3D bioprinting has made significant advances in creating engineered functional tissues, limitations exist when printing living cells with soft biomaterials such as alginate [11,12]. Additionally, unmodified alginate is a bioinert material that requires modifications to allow for cell adhesion [3,13].

RGD (arginine-glycine-aspartate) is a tripeptide sequence that naturally exists in extracellular matrix proteins such as fibronectin and laminin [14,15]. The RGD sequence is crucial to cell adhesion and proliferation as it binds cell-surface integrin receptors to form focal adhesions [16]. Native alginate can be chemically modified to include RGD sequences to promote cell adhesion [16,17,18,19]. A previous study demonstrated that RGD-coupled alginate enhanced cell adhesion, spreading, and differentiation of myoblast cells in hydrogels [17].

Freeform reversible embedding of suspended hydrogels (FRESH) was recently developed to allow for bio-ink extrusion within a thermo-reversible support bath [11,20]. This support bath is composed of gelatin microparticles in a slurry consistency, can provide mechanical support during printing and cross-linking, and is removable at 37 °C. FRESH printing of soft biomaterials such as alginate mitigates the effects of gravity by embedding bio-inks in the aqueous phase of the bath to allow for freeform printing of unsupported structures [20]. Although FRESH printing of native alginates has been reported, the printability and physical properties of RGD-modified alginates printed using the FRESH method have not been studied. The physical properties of RGD-modified alginates are different from those of native alginates [16,19], as with any type of modification, but the impact of this change is unclear in extrusion-based 3D printing using FRESH. Lastly, bioactivity such as cell viability, proliferation, and morphology using the RGD-modified alginate after extrusion-based 3D printing in FRESH remains unclear. Thus, we sought to characterize the physical properties and cellular proliferation of native and RGD-modified alginate after extrusion-based 3D printing in FRESH.

## 2. Materials and Methods

### 2.1. Cell Culture of Adult Human Dermal Fibroblasts

Adult human dermal fibroblasts (HDFa, catalog no. C0135C, ThermoFisher Scientific, Waltham, MA, USA) were cryopreserved as primary cultures. When needed, they were thawed and immediately plated in fibroblast growth media which consisted of human fibroblast expansion basal medium (catalog no. M106500, Gibco, Waltham, MA, USA), 2% (*v*/*v*) low serum growth supplement (catalog no. S00310, ThermoFisher Scientific, Waltham, MA, USA), and 1% (*v*/*v*) penicillin-streptomycin (catalog no. 15140148, ThermoFisher Scientific, Waltham, MA, USA). Cells were cultured in a cell culture incubator (catalog no. 50116047, Thermo Scientific, Waltham, MA, USA) and set to 5% CO_2_ at 37 °C. The media was changed every other day, and the cells were passaged when confluent.

### 2.2. Alginate Bio-Ink Preparation

Native sodium alginate (catalog no. ALG, Allevi Inc., Philadelphia, PA, USA, MW = 295,000 g/mol, M:G ratio = 1.29) and RGD-modified sodium alginate (catalog no. 4270425, NovaMatrix, Sandvika, Bærum, Norway, MW = 188,000 g/mol, M:G ratio = 1.22) were first sterilized using ethylene oxide. The RGD-modified sodium alginate had a RGD peptide substitution of 0.255%, and the RGD:alginate ratio was 0.013 µM/mg prior to printing. The sterilized alginates were dissolved in 50 mM HEPES (catalog no. 15630080, Gibco, Waltham, MA, USA) under stirring at 60 °C to create 2% (*w*/*v*) native sodium alginate (2% SA), 5% (*w*/*v*) native sodium alginate (5% SA), 2% (*w*/*v*) RGD-modified sodium alginate (2% RGD-SA), and 5% (*w*/*v*) RGD-modified sodium alginate solutions (5% RGD-SA). When alginates were fully dissolved, the solution was allowed to cool to 37 °C while stirring. The alginate solutions were stored at 4 °C for up to 48 h if not used immediately. When ready for bioprinting, 2 mL of the alginate solutions were loaded into a 5 mL sterile syringe (catalog no. PSYR5, Allevi Inc., Philadelphia, PA, USA) using sterile technique. Using another 5 mL sterile syringe and a syringe coupler (catalog no. SYRCOUP, Allevi Inc., Philadelphia, PA, USA), 0.01% (*w*/*v*) Alcian Blue 8GX (catalog no. J60122.14, Thermo Scientific, Waltham, MA, USA) was finally mixed in the alginate solutions. Alcian Blue allowed for visualization of the printed constructs prior to cross-linking to avoid loss of constructs during post-print processing. The final alginate solutions were kept in one of the 5 mL sterile syringes, and the other syringe was discarded. Air was removed from the printing syringe by centrifuging the printing syringe at 300× *g* for 1 min, inverting the syringe without introducing air into the alginate solution, and extruding the air from the printing syringe. The syringe was covered with a syringe cap (catalog no. SYRCAP, Allevi Inc., Philadelphia, PA, USA) and stored at 4 °C until ready for printing.

To print cell-laden alginates, HDFa cells of desired concentration were suspended in 10 μL of fibroblast growth media described above. The alginate solutions were warmed to 37 °C prior to cell mixing. This concentrated HDFa mixture was added into a 5 mL sterile syringe with 245 μL of air-free alginate solution using a P20 pipette. Another 5 mL syringe with 245 μL of air-free alginate solution was also prepared and intended for mixing only. Using the syringe coupler, HDFa cells were homogenously suspended in the alginate solutions by moving the plunger back and forth between two sterile, air-free 5 mL syringes. This allowed for mixing without reintroducing air into the solution. After thorough mixing, the cell-laden alginate solutions were kept in one syringe to be used immediately for bioprinting.

To cross-link alginates, CaCl_2_ solution was used. 100 mM sterile CaCl_2_ cross-linking solution was made by dissolving CaCl_2_ (catalog no. CACL, Allevi Inc., Philadelphia, PA, USA) in 50 mM HEPES. This cross-linking solution was then stored in 4 °C for up to 48 h if not used immediately.

### 2.3. FRESH Support Bath Preparation

FRESH support bath is a slurry of spherical gelatin microparticles with an average diameter of 25 μm and reduced polydispersity generated via coacervation. FRESH functions as a thermo-reversible support bath for bioprinting of bio-inks with gelation mechanisms orthogonal to gelatin [11]. The FRESH support bath was prepared for bioprinting according to the recommended protocol with some modifications [11]. Briefly, 20 mg of sterile CaCl_2_ was dissolved in 20 mL of 50 mM HEPES to generate a 0.1% (*w*/*v*) CaCl_2_ solution. This CaCl_2_ solution was then transferred to a 50 mL microcentrifuge conical tube containing 1g of sterile LifeSupport (catalog no. LIFES, Allevi, Philadelphia, PA, USA). Specifically, the LifeSupport powder is composed of dehydrated gelatin microparticles of defined size and shape. The LifeSupport powder was first rehydrated by dissolving the powder in the CaCl_2_ solution by vigorous mixing using a spatula for 1 min to ensure all powder was fully resuspended. The LifeSupport was then kept at 4 °C undisturbed to fully rehydrate. Then the LifeSupport solution was centrifuged at 2000× *g* for 7 min until LifeSupport was compacted, and the resulting supernatant was discarded. The resulting FRESH support bath was transported to well plates for 3D printing. All support baths were used within 12 h.

### 2.4. Extrusion-Based Bioprinting

Allevi 3 Bioprinter (Allevi, Inc., Philadelphia, PA, USA) was used to bioprint alginates in FRESH. The pre-loaded alginate syringe, prepared as described above, was attached to one of the extruders with a 30G, 0.25 in plastic tipped needle attached. After automatic calibration according to the standard protocol, pressure was applied to test extrusion until alginate was dispensed. The extrusion pressures for 2% SA, 5% SA, 2% RGD-SA, and 5% RGD-SA were 7.5 psi, 15 psi, 7.5 psi, and 15 psi, respectively. For construct fidelity testing, 3 mm × 3 mm × 3 mm open single filament boxes were printed. For cell viability and proliferation assessment, 2 mm cylinders of 4 mm in diameters were printed with 100,000 cells/mL in concentration. For cell morphology evaluation, 1 mm cylinders with a 6 mm diameter were printed with 1 million cells/mL in concentration. Cell-laden constructs had an infill grid pattern with an infill distance of 0.16 mm and a layer height of 0.15 mm. All constructs were completed at 22 °C with a print speed of 6 mm/s in 12-well plates (catalog no. 351143, Corning Inc., Corning, NY, USA).

After printing, all constructs were cross-linked with 1 mL of cross-linking CaCl_2_ solution for 30 min at 37 °C. Then, the cross-linking solution was removed, and constructs were washed twice with 1 mL of 50 mM HEPES at 22 °C. Constructs were again cross-linked again with 1 mL of cross-linking solution for another 30 min at 37 °C and washed twice with 50 mM HEPES at 22 °C. Constructs were cross-linked a final time with 1 mL of cross-linking solution for 7 min at 37 °C. Constructs were washed one last time and then cultured in 2 mL of fibroblast growth media in a 5% CO_2_ 37 °C cell culture incubator.

### 2.5. Cell Culture of 3D-Printed Construct

3D-printed constructs were cultured in the fibroblast growth media in a cell culture incubator (catalog no. 50116047, Thermo Scientific, Waltham, MA, USA) set to 5% CO_2_ at 37 °C for up to 28 days. Three days after bioprinting, the growth media was carefully removed using a micropipette without disturbing the constructs. Thereafter, 2 mL of new growth media was added. The media was changed every 3 days until the final timepoint was reached.

### 2.6. Mechanical Tensile Testing

A two-piece mold was 3D-printed in UMA 90 using a 3D Carbon Printer (Carbon, Redwood City, CA, USA). Alginate solutions prepared as described above were cast in the mold to yield alginate samples with ASTM D412-C specified dimensions (Figure 1A,B). Cross-linking CaCl_2_ solution was then sprayed onto the alginate once every 10 min for 30 min. Next, the alginate-containing molds were placed in cross-linking CaCl_2_ solution for 12 h at room temperature to complete cross-linking. Alginate specimens were removed from their molds immediately before analysis with a tensile tester. The alginate specimens were affixed to an Instron 5565 Microtester (Norwood, MA, USA) equipped with a 100 N load cell (Figure 1C). The Instron then tensioned the alginate specimens three times at 5 mm/min until 3 mm of displacement was achieved. Following the third measurement up to 3 mm of displacement, each alginate specimen was tensioned at 5 mm/min until failure. For each alginate specimen (2% SA, 5% SA, 2% RGD-SA, and 5% RGD-SA), the Young’s modulus was calculated from the slope of the linear portion of the stress-strain curve.

### 2.7. Swelling Properties

Alginate strips after cross-linking as described above were carefully blotted with filter paper to remove any excess liquid from the surface. Baseline weight was obtained. The alginate strips were placed in fibroblast growth media in a 37 °C incubator. At 1, 2, 3, 4, 6, 8, 20, and 24 h after incubation, the alginate strips were removed from the growth media, blotted, and weighed again. The swelling ratio was then determined using the following equation:$$Swelling\;Ratio\;\left(\%\right)=\frac{W_{t}-W_{0}}{W_{0}}\times 100$$
where $W_{t}$ is the weight at each time point, and $W_{0}$ is the baseline weight of each sample.

For each alginate, 12 independent samples were used.

### 2.8. Construct Fidelity

To determine the printed constructs’ fidelity, 3 mm × 3 mm × 3 mm open single filament boxes without infill were 3D bio-printed in FRESH and cross-linked. Immediately upon cross-linking completion, we performed fidelity assessment while the constructs were submerged in the cross-linking CaCl_2_ solution. The largest height of the box was measured. The wall thickness was also measured to assess the diameter of a single filament of alginates. For each alginate, 9 independent samples were used.

### 2.9. Cell Viability and Proliferation Assays

To assess cell viability after bioprinting with each alginate bio-ink, a fragment of each printed construct was taken 1, 5, 7, 14, and 28 days after printing. The construct fragments were stained with live/dead viability/cytotoxicity kit for mammalian cells (catalog no. L3224, Invitrogen, Carlsbad, CA, USA) according to the standard protocol. After staining for 30 min, the construct fragments were placed on a concave glass microscope slide (catalog no. 7104, Chang Bioscience, Fremont, CA, USA) and imaged with a Leica DMi8 microscope (Leica Microsystems, Wetzlar, Hesse, Germany). Images were further analyzed using ImageJ (National Institutes of Health, Bethesda, MD, USA). For each alginate at each time point, 6 samples were measured.

To assess cellular proliferation, proliferation assays were performed 1, 5, 7, 14, and 28 days after printing. At each time point, alginate constructs were moved to a new 12-well plate and submerged in 2 mL of fibroblast growth media. Next, 200 µL of Cell Counting Kit-8 (CCK8, catalog no. CK0411, Dojindo Molecular Technologies, Rockville, MD, USA) was added to each well. The constructs were incubated at 37 °C for 3 h. Using BioTek Synergy 2 (Agilent Technologies, Santa Clara, CA, USA) and BioTek Gen 5 Software (Agilent Technologies, Santa Clara, CA, USA), the optical density at 450 nm (OD_450_) of each well was obtained.

### 2.10. Cell Morphology

To assess cellular morphology, constructs were taken 1 and 7 days after printing for staining. The printed constructs were first fixed at room temperature using 4% paraformaldehyde (catalog no. J19943.K2, Thermo Fisher, Waltham, MA, USA) and 1% (*w*/*v*) CaCl_2_ in 50 mM HEPES for 30 min. The constructs were then washed with 50 mM HEPES. Next, the constructs were permeabilized by treating them with 0.1% (*v*/*v*) Triton X-100 (catalog no. X-100, Sigma-Aldrich, St. Louis, MO, USA) and 1% (*w*/*v*) CalCl_2_ in 50 mM HEPES for 30 min at room temperature followed by another wash with 50 mM HEPES. The constructs were then blocked in 1% (*v*/*v*) BSA (catalog no. A1595, Sigma-Aldrich, St. Louis, MO, USA) and 1% (*w*/*v*) CaCl_2_ in 50 mM HEPES for 1 h at room temperature. After that, the constructs were washed in 50 mM HEPES before incubating overnight at 4 °C with (1:200) anti-rabbit actin antibody (catalog no. PIMA532479, Thermo Fisher, Waltham, MA, USA), (1:200) anti-mouse integrin beta 1/CD29 antibody (catalog no. MAB1778, Thermo Fisher, Waltham, MA, USA), and 1% (*w*/*v*) CaCl_2_ in 50 mM HEPES. After washing the constructs in 50 mM HEPES, the constructs were then incubated for 2 h at room temperature in (1:200) Alexa Fluor 594 Goat Anti-Rabbit IgG H&L (catalog no. ab150080, Abcam, Waltham, MA, USA), (1:200) Alexa Fluor 488 Goat Anti-Mouse IgG H&L (catalog no. ab150113, Abcam, Waltham, MA, USA), and 1% (*w*/*v*) CaCl_2_ in 50 mM HEPES. Following another wash in 50 mM HEPES, DAPI (catalog no. NC9524612, Fisher Scientific, Waltham, MA, USA) was added to stain the constructs. The stained constructs were then imaged using a Zeiss LSM 780 inverted confocal microscope and Zeiss Xen software at 20×/0.5 (ZEISS Microscopy, Jena, Germany).

### 2.11. Statistical Analysis

Continuous variables were reported as mean ± standard deviation unless specified otherwise. Sample variance was assessed using F-test. To compare each metric (at each time point if applicable) among the different alginate bio-inks, analysis of variance (ANOVA) with Tukey post-hoc correction was performed. A *p* value of less than 0.05 was considered statistically significant.

## 3. Results

### 3.1. Mechanical Properties of Alginates

Native alginates and RGD-modified alginates at different concentrations demonstrated a broad range of mechanical properties (*p* < 0.0001, Figure 2A). In general, alginates at higher concentrations were stiffer, with the 2% vs. 5% SA having a Young’s modulus of 492.3 ± 13.0 kPa vs. 1816.8 ± 61.0 kPa (*p* < 0.0001), respectively; and 2% vs. 5% RGD-SA with Young’s modulus measuring 531.8 ± 26.8 kPa vs. 1767.0 ± 57.5 kPa (*p* < 0.0001), respectively. RGD modification did not significantly alter alginates’ stiffness. The force vs. displacement plots are shown in Appendix A.

### 3.2. Swelling Ratio of Alginates

Swelling ratios differed significantly between the alginates at each time point from after 1 h (*p* < 0.0001), 2 h (*p* = 0.0002), 3 h (*p* = 0.0004), 4 h (*p* < 0.0001), 6 h (*p* < 0.0001), 8 h (*p* < 0.0001), 20 h (*p* < 0.0001), and 24 h (*p* < 0.0001) of incubation (Figure 2B). Significant swelling appeared within the first hour of incubation. Specifically, 5% RGD-SA compared to 2% SA as well as compared to 2% RGD-SA demonstrated a significantly higher swelling ratio at all time points (*p* < 0.01). Furthermore, 5% RGD-SA maintained a significantly higher swelling ratio compared to 5% SA starting at 2 h after incubation (*p* < 0.05). There was no statistically significant difference in swelling ratio between 2% SA and 2% RGD-SA at any time point.

### 3.3. Alginate Bioprinting Fidelity

3D bioprinting using alginates in FRESH resulted in generally excellent fidelity in shape, and constructs maintained a squared box geometry in 3D (Figure 3A–D). The largest height of the box for 2% SA, 5% SA, 2% RGD-SA, and 5% RGD-SA were 3.7 ± 0.5 mm, 3.2 ± 0.1 mm, 3.3 ± 0.2 mm, and 3.1 ± 0.1 mm, respectively (*p* = 0.0001, Figure 3E). Specifically, 2% SA constructs were larger compared to constructs printed with 2% RGD-SA (*p* = 0.02), 5% SA (*p* = 0.002), and 5% RGD-SA (*p* = 0.001). Compared to the desired construct size, the final 3D constructs using 2% SA, 5% SA, 2% RGD-SA, and 5% RGD-SA were 23.7 ± 16.7%, 6.3 ± 3.3%, 10.9 ± 6.5%, and 2.6 ± 2.2% larger in maximal height (*p* = 0.0001). In terms of filament sizes, constructs printed with 2% SA, 5% SA, 2% RGD-SA, and 5% RGD-SA had wall thickness measured 1.8 ± 0.8 mm, 1.0 ± 0.1 mm, 1.3 ± 0.4 mm, and 0.8 ± 0.1 mm, respectively (*p* = 0.0004, Figure 3E).

### 3.4. HDFa Cell Viability in 3D Bio-Printed Alginates

3D bioprinting using HDFa cell-laden alginates as bio-ink was successful (Figure 4A,B and Appendix A). HDFa cell viability at day 1 after bioprinting remained high with an average viability of 88.3 ± 1.0%, 85.6 ± 3.5%, 84.1 ± 3.5%, and 84.0 ± 1.4% using 2% SA, 5% SA, 2% RGD-SA, and 5% RGD-SA, respectively (*p* = 0.04). Throughout the 28-day incubation period (Figure 4C), cell viability in 2% SA remained the lowest compared to 2% RGD-SA and 5% RGD-SA at day 7 (69.9 ± 2.8% vs. 81.3 ± 3.4% and 82.8 ± 4.4%, *p* = 0.01 and 0.004), day 14 (66.6 ± 3.3% vs. 85.9 ± 11.3% and 81.1 ± 6.5%, *p* = 0.01 and 0.03), and day 28 (56.2 ± 6.3% vs. 84.3 ± 3.9% and 80.1 ± 6.3%, *p* = 0.001 and 0.001). At day 28, cells printed in 5% SA also demonstrated significantly lowered viability compared to 2% RGD-SA (69.4 ± 6.1% vs. 84.3 ± 3.9%, *p* = 0.02), though still higher than 2% SA (*p* = 0.01). Additionally, cell viability in SA regardless of the concentration continued to decrease over time. There was no difference in cell viability printed in 2% or 5% RGD-SA throughout the 28-day incubation period and the proportion of viable cells in RGD-SA remained stable over time.

HDFa cell proliferation measured after bioprinting in alginates varied over time (Figure 4D). Specifically, 5% RGD-SA consistently supported the highest cell proliferation compared to 2% RGD-SA at day 5 (0.19 ± 0.03 vs. 0.14 ± 0.01, *p* = 0.001) compared to 2% SA (0.24 ± 0.05 vs. 0.18 ± 0.01, *p* = 0.04) and 5% SA (0.24 ± 0.05 vs. 0.17 ± 0.01, *p* = 0.004) at day 28, respectively. Furthermore, both RGD-SAs supported an increase in cell growth over 28 days, but 5% RGD-SA demonstrated better cell proliferation over time compared to 2% RGD-SA.

### 3.5. HDFa Cell Morphology in 3D Bio-Printed Alginates

HDFa cells were well-visualized in the alginates after 3D bioprinting (Figure 5). There were no significant visual differences in cell morphology on day 1. However, slightly enhanced but faint integrin signals were observed throughout cell cytoplasm in cells printed in 2% and 5% RGD-SAs compared to those printed in native alginates (Appendix A). After 7 days of culture, cell morphology remained similar in different alginates. However, actin signals were more pronounced compared to cells 1 day after bioprinting (Appendix A). The integrin signal was only observed in the cells printed in 5% RGD-SA. Area of integrin aggregation and clustering around the cell surface was also observed (Appendix A). However, the integrin signal disappeared in cells printed in 2% RGD-SA, similar to cells printed in native alginates after 7 days of culture.

## 4. Discussion

FRESH 3D bioprinting is a significant advancement for the bio-fabrication of soft materials such as alginates. Although FRESH bioprinting native alginates and physical properties of RGD-modified alginates have been reported independently, the physical properties of 3D bio-printed RGD-modified alginates using the FRESH technique have not been reported previously. This represents a knowledge gap that requires further investigation. Active research is being conducted to modify alginates and printing methodologies to improve scaffold physical properties, printability, and bioactivity [5]. In this study, we characterized the physical properties, printability, and cellular proliferation of native and RGD-modified alginate after extrusion-based 3D bioprinting in FRESH. Given that RGD-modified alginates are attractive biomaterials for cell growth and differentiation [16], and that FRESH 3D bioprinting is a popular printing technique to generate scaffolds in complex geometry with high precision [11], we believe this study may have an important impact in the field of 3D bioprinting to facilitate further biomedical research in biomaterials and tissue engineering.

In terms of physical properties, bio-inks with a higher concentration of alginates were found to be associated with a higher Young’s Modulus. This was in accordance with previous literature published on native alginates [16,17,21]. Previous literature also reported a significant effect on mechanical properties from adjusting the molecular weight of alginate [22,23,24]. Interestingly, the mechanical property can be further tuned by modifying alginates with RGD. We also found that a higher concentration of alginates was associated with a higher swelling ratio for 5% RGD-SA, which demonstrated the highest swelling ratio amongst the four alginates investigated in this study. Water absorption is an important feature for hydrogels and tissue engineering scaffolds as it reflects the scaffold’s ability to absorb body fluids and transport water and nutrients [25]. All alginates after FRESH 3D bioprinting had relatively high swelling ratios. The varying swelling ratios observed in this study may be due to the different degrees of crosslinking because of different alginate concentrations and the addition of RGD motifs [25].

In terms of bio-ink printability and construct fidelity, 5% RGD-SA demonstrated the best fidelity with highest precision amongst the different alginates used in this study. Printability of a bio-ink is impacted by the mechanical stability of the first layer, rheological properties, as well as cross-linking mechanisms [5]. For lower concentration of alginate, the viscosity was lower, and the bio-ink was more spread out after extrusion. This further impacts the rheology properties of alginates, as reflected by the higher extrusion pressure required to print alginates at higher concentrations. The width of the printed filament was the finest using 5% RGD-SA, suggesting the best construct fidelity amongst the alginates tested in this study. In fact, a previous study has shown slightly increased viscosity of RGD-SA compared to native SA of the same concentration [26]. This slight change of viscosity, though not significant enough to affect extrusion pressure required for bioprinting, allowed for finer filament deposition.

Throughout the printing process and the post-print processing, many factors can impact cell viability. These include but are not limited to shear stress during extrusion-based bioprinting, fabrication time, gelling condition, cross-linking methodology, and duration [10]. In this study, we controlled for variables to isolate the impact of bio-ink alone on cell viability. We showed that cell viability in all alginates remained high until 7 days post-bioprinting when RGD-modified alginates demonstrated superior viability. We hypothesize that the presence of RGD provides a favorable 3D micro-environment to allow for cell adhesion and further facilitates enhanced cell growth and proliferation over an extended period. Our study demonstrated significantly augmented cell proliferation associated with 5% RGD-modified alginates compared to native alginates at the 28-day time point. Prior studies showed that RGD-modified alginates were able to support cell proliferation to maintain 80% of viability for at least 2 days [27]. Cell migration and organization have also been observed [26]. The findings from this study again suggest that the presence of RGD motif in alginates not only helps maintain cell viability but also supports cell growth and proliferation. Although the cell-matrix interaction gets less important as time passes as cell-cell contacts and interactions develop, the initial 3D micro-environment when the cells were printed can be a critical factor to determine cell fate [28]. In our study, we found that cell reorganization with actin polymerization was prominent 7 days after bioprinting across different alginates. Since prior literature observed cytoskeleton remodeling after cells being exposed to shear stress [29,30], we hypothesize that the extrusion-based 3D bioprinting process can activate cytoskeleton. Interestingly, integrin was initially present in both 2% and 5% RGD-modified alginates 1 day after bioprinting, but after 7 days of culture, integrin was only observed on the cell surface in those printed in 5% RGD-SA. These cells also demonstrated increased integrin density and aggregation after 7 days of culture. This might explain why cell proliferation was significantly improved when cells were printed in 5% RGD-SA as integrin binding to the extracellular matrix can contribute to the cell’s decision to proliferate, migrate, or die [31]. The RGD peptide density may also impact cell proliferation and viability, given that we showed cells printed in 2% RGD-SA lost integrin after 7 days culture, whereas cells printed in 5% RGD-SA presented enhanced integrin with focal aggregates after 7 days of culture. We hypothesize that with lower RGD peptide density, integrin clusters were unable to form, thereby affecting integrin-extracellular matrix bonds, adhesive force distribution, cell signaling systems, migration, and proliferation [32,33,34].

## 5. Conclusions

In conclusion, we described tunable physical properties of native and RGD-modified alginates after FRESH 3D bioprinting. Sodium alginate with RGD modification, especially at a high concentration, was associated with greatly improved cell viability and integrin clustering, which further enhanced cell proliferation. A few limitations should be mentioned. While construct fidelity was assessed immediately upon cross-linking completion, this should be more comprehensively evaluated over time and in physiologically relevant conditions. Future studies should also analyze and quantify RGD substitution in alginates after printing and in cell culture to confirm the presence or absence of RGD motif release from the cell-alginate matrix during cell culture and incubation. Further analysis should also be conducted to evaluate the mechanism of cytoskeleton activation upon 3D bioprinting.

## Figures and Tables

**Figure 1 bioengineering-09-00807-f001:**
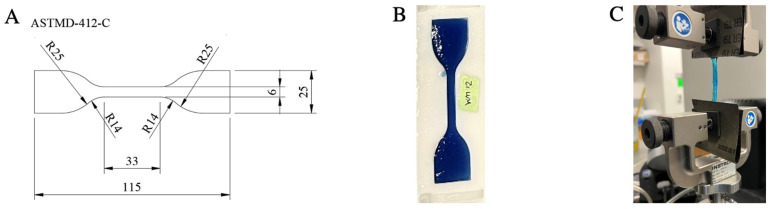
Alginate specimens molded for mechanical tensile testing and swelling ratio assessment. (**A**) ASTMD-412-C specified dimensions. All units are in millimeters. (**B**) 3D-printed molds were used to cast alginate specimens with ASTM D412-C specified dimensions. (**C**) The alginate specimens were secured to an Instron tester for Young’s modulus measurement.

**Figure 2 bioengineering-09-00807-f002:**
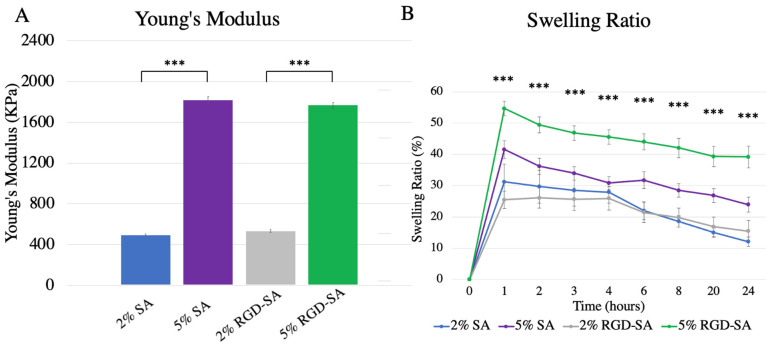
Physical properties of alginates. (**A**) Mechanical tensile testing performed for four different alginates demonstrated significantly different Young’s Modulus (*p* < 0.0001) with 5% (*w*/*v*) sodium alginates, with or without RGD modification, demonstrating statistically significantly higher Young’s Modulus compared to 2% (*w*/*v*) alginates with or without RGD modification. (**B**) Swelling ratio increased within the first hour for all alginates, but steadily decreased over the course of 24 h. At each time point, swelling ratios of different alginates were significantly different from each other with 5% RGD-SA demonstrating the highest swelling ratio amongst all alginates. *** denotes *p* < 0.001. SA: sodium alginate; RGD-SA: RGD-modified sodium alginate. Error bars represent standard error.

**Figure 3 bioengineering-09-00807-f003:**
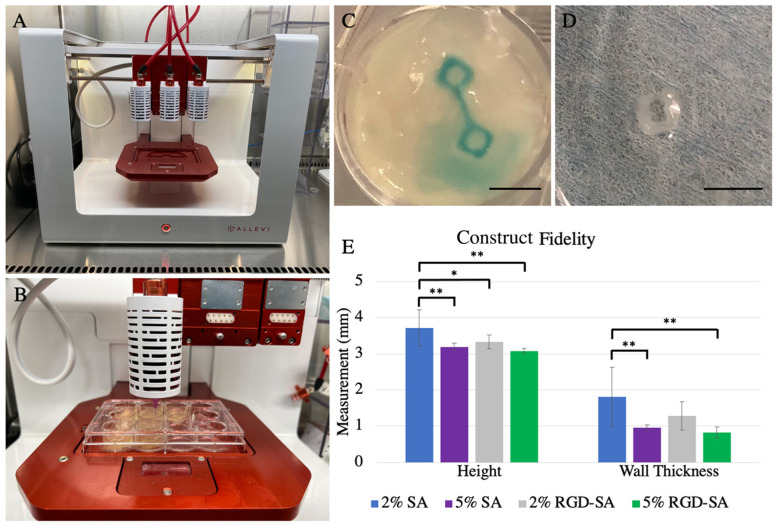
3D bioprinting using the Allevi 3 bioprinter and alginates as bio-ink in freeform reversible embedding of suspended hydrogels (FRESH) for construct fidelity testing (**A**) Photograph of the Allevi 3 3D bioprinter with all 3 extruders attached. (**B**) Demonstration of 3D bioprinting of alginates in FRESH. (**C**) Open, single filament alginate boxes of 3 mm × 3 mm × 3 mm in size were printed in FRESH prior to cross-linking. Scale bar = 5 mm. (**D**) An exemplary open, single filament alginate box after cross-linking. Scale bar = 5 mm. (**E**) Construct fidelity analysis compared box maximal height and wall thickness printed using four different alginates. Mixture of 2% (*w*/*v*) sodium alginate (SA) demonstrated the largest box height and wall thickness, whereas 5% (*w*/*v*) SA with or without RGD modification demonstrated smaller box height and wall thickness. * denotes *p* < 0.05; ** denotes *p* < 0.01. SA: sodium alginate; RGD-SA: RGD-modified sodium alginate. Error bars represent standard deviation.

**Figure 4 bioengineering-09-00807-f004:**
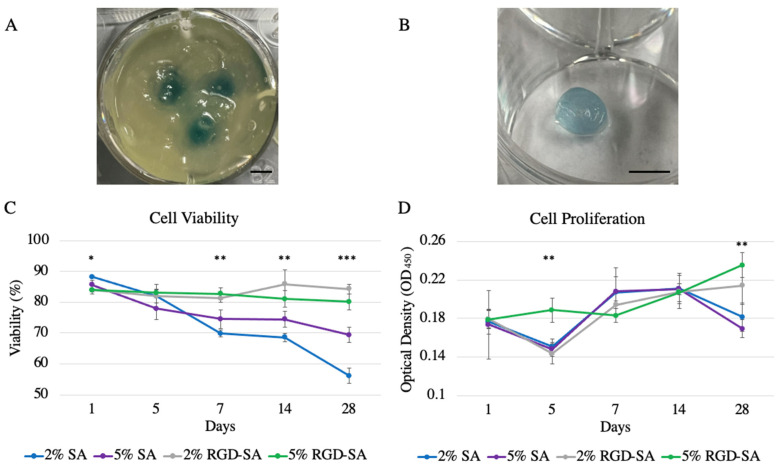
Adult human dermal fibroblast cell viability and proliferation after 3D bioprinting in different alginate bio-inks over 28 days. (**A**) 6 mm cylinders were 3D printed with adult human dermal fibroblast-laden alginates in freeform reversible embedding of suspended hydrogels (FRESH). Scale bar = 5 mm. (**B**) After FRESH removal and cross-linking, the 3D printed cylinder was successfully harvested for further cell viability and proliferation testing. Scale bar = 5 mm. (**C**) Cell viability in 2% (*w*/*v*) sodium alginate (SA) demonstrated the lowest cell viability compared to RGD-modified SA (RGD-SA) after 7 days. Cell viability in 2% SA continued to decrease over time. Cells printed in RGD-SA remained viable over time. (**D**) Cells after 3D bioprinting in different alginates demonstrated varied proliferation. 5% RGD-SA demonstrated the highest cell proliferation compared to 2% RGD-SA at day 5 and compared to SA at day 28. Cells were able to successfully proliferate in RGD-SA with 5% RGD-SA supported better cell proliferation compared to 2% RGD-SA. * denotes *p* < 0.05; ** denotes *p* < 0.01; *** denotes *p* < 0.001. Error bars represent standard error.

**Figure 5 bioengineering-09-00807-f005:**
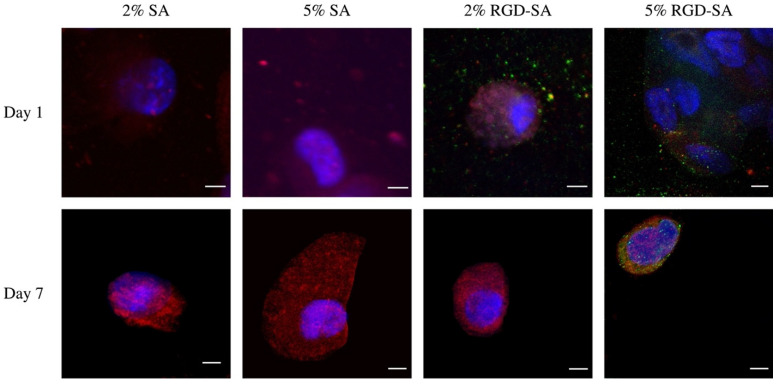
Cell morphology 1 and 7 days after bioprinting in alginates. Cells appeared similar in appearance 1 day after bioprinting, but only RGD-modified alginates (RGd-SAs) showed integrin signals. However, 7 days after bioprinting, only cells printed in 5% RGD-SA demonstrated integrin signals with clustering. Cell morphologies were otherwise similar in different alginates 7 days after bioprinting. SA: sodium alginate. Scale bar = 5 μm.

## Data Availability

Data will be made available upon reasonable request to the corresponding author.

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
