# Peer review of "Three-Dimensional Bioprinting with Alginate by Freeform Reversible Embedding of Suspended Hydrogels with Tunable Physical Properties and Cell Proliferation"

_bioengineering, 2022, doi:10.3390/bioengineering9120807_

Round 1

Reviewer 1 Report

Tissue engineering scaffold is a bridge to repair diseased tissues. Exploring the effects of cell adhesion, differentiation and proliferation in scaffold materials is of great significance for understanding and regulating the function of tissue engineering scaffold. The printability of alginate hydrogel modified by arginine glycine aspartate and its ability to adhere, differentiate and proliferate stem cells were studied by extrusion molding technology. The research method of this paper is reasonable, and the expected research results have been obtained. It has important application value for the development of tissue engineering scaffold materials and can be published.

Author Response

Thank you for your prompt and thorough review of our manuscript. We made appropriate changes to address the language and style throughout the manuscript.

Reviewer 2 Report

1. Alginate-RGD has been widely applied in 3DP bioink, and FRESH method is not novel at all. The specific contribution of this study is not clear.

2. Does the Alginate-RGD bioink bring about better printing efficiency or resolution? Is it necessary to adjust the printing parameters due to RGD modification?

3. What are the RGD amounts in bioink before printing, after printing, and in the cell culture? What is the RGD density on the cell-alginate interface?  Is the RDG motif released from cell-alginate matrix after the incubation for a certain period?

4. It is interesting that the cytoskeleton would be more activated by applying Alginate-RGD bioink and FRESH printing. However, the mechanism and duration is not clearly described in this manuscript. If conventional extrusion bioprinting is applied, is the cytoskeleton also activated?

Author Response

  1. Alginate-RGD has been widely applied in 3DP bioink, and FRESH method is not novel at all. The specific contribution of this study is not clear.

Thank you for your important comment. We agree with your assessment. Though RGD-modified alginates and FRESH bioprinting have been described separately, the use of RGD-modified alginates in FRESH bioprinting has not been described. This study focused on FRESH 3D-printed RGD-modified alginates specifically and aimed to characterize the physical properties and the associated cellular proliferation.

  1. Does the Alginate-RGD bioink bring about better printing efficiency or resolution? Is it necessary to adjust the printing parameters due to RGD modification?

We believe RGD-modified alginates are associated with better printing resolution compared to un-modified alginates (see Figure 2). We did not need to adjust printing parameters due to RGD modification. The extrusion pressures for 2% SA, 5% SA, 2% RGD-SA, and 5% RGD-SA were 7.5 psi, 15 psi, 7.5 psi, and 15 psi, respectively, all printed at 22°C with a print speed of 6mm/s.

  1. What are the RGD amounts in bioink before printing, after printing, and in the cell culture? What is the RGD density on the cell-alginate interface?  Is the RDG motif released from cell-alginate matrix after the incubation for a certain period?

Thank you very much for your question. Prior to bioprinting, the degree of the RGD peptide substitution was 0.255% and the RGD:alginate ratio was 0.013 mM/mg. For the purpose of this study, we did not characterize the RGD concentration after printing and in the cell culture, but we are working on answering this question as part of our ongoing studies. We included these comments in the Methods and Discussion sections.

  1. It is interesting that the cytoskeleton would be more activated by applying Alginate-RGD bioink and FRESH printing. However, the mechanism and duration is not clearly described in this manuscript. If conventional extrusion bioprinting is applied, is the cytoskeleton also activated?

This is a very interesting question – thank you. There are a few prior papers described cytoskeleton activation and remodeling after cells were exposed to shear stress. We suspect that this might be part of the reasons why the cytoskeletons were more activated after printing. We included these comments in the Discussion section along with the new references in the revised manuscript.

Reviewer 3 Report

It`s a good try to evaluate the possibility of 3D print for RGD-SA, as some papers have shown its good cell adhesion and spreading performance. The crosslinking treatment can enhance the mechanical strength and keep its shape during the 3D printing. In my opinion, this paper is good enough to pubilish. Besides, just to discuss a little after the publication furthermore, as some papers shows that the RGD could fall off the material, if it is used as a real material, what`s your opinion on its further modifacation? This question is irrelevant with my decision, it`s just a scientific disscusion, thank you.

Author Response

Thank you very much for your insightful comment. Prior to bioprinting, the degree of the RGD peptide substitution was 0.255% and the RGD:alginate ratio was 0.013 mM/mg. For the purpose of this study, we did not characterize the RGD concentration after printing and in the cell culture, but we are working on answering this question as part of our ongoing studies. We included these comments in the Methods and Discussion sections.

Reviewer 4 Report

The work by Zhu et al. provides a route for the implementation of FRESH in 3D extrusion bioprinting applications. However, prior to publication, the authors need to address some issues:

1. The authors claimed that air bubbles were removed before extrusion but failed to explain how.

2. Please provide more details on how LifeSupport is composed.

3. By prints you meant constructs? I suggest changing it to construct as it is the concept most commonly used in the literature. Then the authors refer to them as structures. Please use just one and stick to it throughout the document.

4. Why did the authors use strips instead of the conventional geometries required by the ASTM standards for tensile strength testing? This is risky as stress concentrations may arise and alter the results significantly.

5. Please include the complete stress vs. strain plot to make sure that the results are consistent and repeatable.

6. The details of the imaging process by confocal microscopy are incomplete. Please include NA, Objective, Ex/Em, etc. 

7. Shape fidelity of constructs needs to be also measured in time and under physiological or relevant conditions.

8. The proliferation results are not well explained and rationalized in light of the viability results by the authors.

9. Please include a conclusions section with the major findings and limitations of the study.

Author Response

  1. The authors claimed that air bubbles were removed before extrusion but failed to explain how.

Thank you for this important comment. We added additional information regarding this in the Methods section to provide more details.

  1. Please provide more details on how LifeSupport is composed.

Thank you for your suggestion. We added additional information about the LifeSupport bath in the Methods section.

  1. By prints you meant constructs? I suggest changing it to construct as it is the concept most commonly used in the literature. Then the authors refer to them as structures. Please use just one and stick to it throughout the document.

Thank you for your recommendation. We modified this throughout the manuscript and used “construct” to be consistent.

  1. Why did the authors use strips instead of the conventional geometries required by the ASTM standards for tensile strength testing? This is risky as stress concentrations may arise and alter the results significantly.

Thank you for your recommendation. We modified our testing setup to comply with the ASTM standards for tensile strength testing. We included the updated methods, results, and data interpretation throughout the manuscript.

  1. Please include the complete stress vs. strain plot to make sure that the results are consistent and repeatable.

Thank you for your suggestion. We included the complete stress/strain curves in our supplemental figures.

  1. The details of the imaging process by confocal microscopy are incomplete. Please include NA, Objective, Ex/Em, etc. 

Thank you for pointing this out. We added additional specifications in the Methods section.

  1. Shape fidelity of constructs needs to be also measured in time and under physiological or relevant conditions.

We agree with your assessment. For the purpose of this paper, we only assessed shape fidelity immediately after cross-linking in cross-linking solutions. As a part of our ongoing research, we will further evaluate shape fidelity over time and under physiological conditions. We added these information in the Discussion section.

  1. The proliferation results are not well explained and rationalized in light of the viability results by the authors.

Thank you for your insightful comment. We added a few more comments on our findings in the Discussion section.

  1. Please include a conclusions section with the major findings and limitations of the study.

Thank you for your recommendation. We further strengthened our revised manuscript by including a limitation paragraph in the Discussion section.

Round 2

Reviewer 2 Report

1.        The authors mentioned that printability was enhanced by RGD-modification. However, the mechanism was not well described. For example, the increase in MW enhanced the printability of alginate would be due to the promotion of intermolecular forces or entanglement interactions which can be found from viscosity or Tg. What is the mechanism of the enhanced printability of RGD-modified alginates in this study? If the RGD-modification enhances the swelling, how does it promote printability?

2.        The cytoskeleton activation and immobilized-RGD quantification were not well identified in this manuscript; however, it is acceptable to describe the related mechanism and carry out analysis in ongoing studies.

Author Response

  1. The authors mentioned that printability was enhanced by RGD-modification. However, the mechanism was not well described. For example, the increase in MW enhanced the printability of alginate would be due to the promotion of intermolecular forces or entanglement interactions which can be found from viscosity or Tg. What is the mechanism of the enhanced printability of RGD-modified alginates in this study? If the RGD-modification enhances the swelling, how does it promote printability?

Thank you very much for your feedback. We believe that swelling differences may be due to different degrees of crosslinking in the four bio-inks used in this study. The enhanced printability of RGD-modified alginates may be due to increased viscosity compared to native alginates. Though crosslinking and viscosity can both impact printability and swelling, we think these are two factors that separately driving the physical properties of alginates that are produced using FRESH printing, which is the unique aspect of the paper. We added our comments about these hypotheses in the Discussion section.

2.        The cytoskeleton activation and immobilized-RGD quantification were not well identified in this manuscript; however, it is acceptable to describe the related mechanism and carry out analysis in ongoing studies.

Thank you for your suggestion. We added this into our limitation/future steps paragraph in the Discussion section.